# Artificial Extracellular Matrices Containing Bioactive Glass Nanoparticles Promote Osteogenic Differentiation in Human Mesenchymal Stem Cells

**DOI:** 10.3390/ijms222312819

**Published:** 2021-11-26

**Authors:** Lysann M. Kroschwald, Felix Allerdt, Anne Bernhardt, Sandra Rother, Kai Zheng, Iram Maqsood, Norbert Halfter, Christiane Heinemann, Stephanie Möller, Matthias Schnabelrauch, Michael C. Hacker, Stefan Rammelt, Aldo R. Boccaccini, Vera Hintze

**Affiliations:** 1Centre for Translational Bone, Joint and Soft Tissue Research, University Hospital “Carl Gustav Carus”, Technische Universität Dresden, Fetscherstraße 74, D-01307 Dresden, Germany; Lysann.Kroschwald@uniklinikum-dresden.de (L.M.K.); Anne.Bernhardt@tu-dresden.de (A.B.); 2Institute of Materials Science, Max Bergmann Center of Biomaterials, TU Dresden, Budapester Straße 27, D-01069 Dresden, Germany; Felix.Allerdt@gmx.de (F.A.); sandra.rother@uks.eu (S.R.); Norbert.Halfter@tu-dresden.de (N.H.); Christiane.Heinemann@tu-dresden.de (C.H.); 3Institute of Biomaterials, University of Erlangen-Nuremberg, D-91058 Erlangen, Germany; kai.zheng@fau.de (K.Z.); aldo.boccaccini@fau.de (A.R.B.); 4Institute for Pharmacy, Pharmaceutical Technology, University Leipzig, D-04317 Leipzig, Germany; irammaqsood2@yahoo.com; 5Riphah Institute of Pharmaceutical Sciences (RIPS), Riphah International University (RIU), Lahore 54000, Pakistan; 6Biomaterials Department, INNOVENT e.V., D-07745 Jena, Germany; sm@innovent-jena.de (S.M.); ms@innovent-jena.de (M.S.); 7Institute of Pharmaceutics and Biopharmaceutics, Heinrich Heine University, D-40225 Düsseldorf, Germany; michael.hacker@hhu.de; 8University Centre for Orthopaedics, Plastic and Trauma Surgery, University Hospital Carl Gustav Carus, D-01307 Dresden, Germany; stefan.rammelt@uniklinikum-dresden.de

**Keywords:** artificial extracellular matrices, bioactive glass nanoparticles, glycosaminoglycans, mesenchymal stem cells, osteogenic differentiation

## Abstract

The present study analyzes the capacity of collagen (coll)/sulfated glycosaminoglycan (sGAG)-based surface coatings containing bioactive glass nanoparticles (BGN) in promoting the osteogenic differentiation of human mesenchymal stroma cells (hMSC). Physicochemical characteristics of these coatings and their effects on proliferation and osteogenic differentiation of hMSC were investigated. BGN were stably incorporated into the artificial extracellular matrices (aECM). Oscillatory rheology showed predominantly elastic, gel-like properties of the coatings. The complex viscosity increased depending on the GAG component and was further elevated by adding BGN. BGN-containing aECM showed a release of silicon ions as well as an uptake of calcium ions. hMSC were able to proliferate on coll and coll/sGAG coatings, while cellular growth was delayed on aECM containing BGN. However, a stimulating effect of BGN on ALP activity and calcium deposition was shown. Furthermore, a synergistic effect of sGAG and BGN was found for some donors. Our findings demonstrated the promising potential of aECM and BGN combinations in promoting bone regeneration. Still, future work is required to further optimize the BGN/aECM combination for increasing its combined osteogenic effect.

## 1. Introduction

There is a high need for bone replacement materials and effective options to promote the bone healing response due to an aging population with an increasing number of multimorbid patients as well as high incidences of delayed or failed integration of bone substitute materials [1]. Considering the limitations of traditional therapies, nanomaterials provide novel strategies for bone regeneration as bone itself is a nanomaterial composed of organic and inorganic components with a hierarchical structure ranging from nanoscale to macroscale [2]. One promising route in the development of novel functional biomaterials is to utilize components of the extracellular matrix (ECM), known to promote the osteogenic response, and thereby mimicking the osteogenic niche. In this context, sulfated glycosaminoglycans (sGAG) and bioactive glass nanoparticles (BGN) are both well recognized for their osteogenic properties [3,4,5,6]. Notably, chemically sulfated hyaluronan (sHA), as part of artificial ECM (aECM), is reported to promote bone regeneration [7,8,9,10] by enhancing multiple functions of mesenchymal stem cells (MSC) and premature osteoblasts [11,12,13,14,15]. This includes cell–matrix interactions, cell-signaling, endocytosis, and osteogenic differentiation.

Specifically, silicon and calcium ions as main inorganic components of BGN are known to be essential natural nutrients responsible for diverse biological processes [16,17]. The release of silicon and, at later time points of incubation, calcium ions was reported for the silicate-based BGN used in this study [18]. Silicon and calcium ions are reported to favor intracellular and extracellular responses, to promote MSC and osteoblast proliferation and differentiation [19,20,21,22,23,24,25] as well as to accelerate bone formation [17,26]. For bone-related applications, silicate-based BGN show unique properties for regeneration and repair due to their intrinsic bioactivity coupled with their small size and high specific surface area [18]. BGN, as nanoscale material, are preferably used as fillers to enhance mineralization, mechanical and osteogenic activities of biopolymeric matrices [27,28], as they usually enhance these properties to a greater extent in comparison to microscale particles [29].

In order to evaluate potential synergistic effects of either CS or sHA in combination with BGN, we investigated, for the first time, the osteogenic capacity of human MSC (hMSC) exposed to collagen-coatings comprising sGAG alone or associated with BGN. These aECM were comprehensively characterized in terms of composition, stability, and morphology as well as viscoelastic properties. Finally, they were evaluated for their effect on proliferation and osteogenic differentiation of hMSC in the presence and absence of dexamethasone (Dex).

## 2. Results

### 2.1. Composition and Stability of aECM Coatings

The composition of aECM coatings was assessed by directly analyzing the collagen and sGAG content in coatings after preparation (Figure 1A–C). The collagen content of aECM directly after preparation was between 410 and 875 µg/cm² (Figure 1A). Initially about 60 µg/cm² CS and 80–85 µg/cm² sHA3 could be electrostatically associated with the aECM with no major impact of BGN (Figure 1B). The presence of collagen and sGAG in the initial coatings was furthermore proven by respective staining (Figure 1C). However, the macroscopic appearance of the coatings differed in such that matrices containing sGAG were rather uniform and homogeneous, irrespective of the presence of BGN. In contrast, matrices with collagen as the only organic component displayed a less uniform distribution on tissue culture polystyrene. Here, the collagen fibrils appeared to accumulate in certain areas, while others remained sparsely coated. This was again irrespective of the presence of BGN.

The stability of the coatings was determined by analyzing the released amount of collagen and sGAG in supernatants (Figure 1D,E) after incubation in PBS at 37 °C. The strongest release of collagen as well as sGAG was detected within the first day, while afterwards the release was markedly diminished (Figure 1D,E). The initial mass composition of the GAG-containing aECM was 86–88% collagen and 12–13% GAG. After one day in PBS at 37 °C this composition changed to 97–98% collagen and 2–3% CS or 88–91% collagen and 9–11% sHA3, respectively. Thus, the CS-containing matrices released 84–88% of their CS content during this time, whereas this was only 35–44% in case of coll/sHA3 matrices. A significantly lower loss of sHA3 was observed in the presence of BGN, whereas there was no impact on CS release. After 13 days, there were still more than three-quarters of the initial collagen content present in aECM (Figure 1A,D). For CS-containing aECM, after 13 days about 87–93% of the initially bound sGAG was lost, while for sHA3-containing aECM it was 40–48% (Figure 1B,E).

Furthermore, aECM coatings were incubated in cell culture medium over 31 days. Calcium concentration was assessed by a colorimetric assay as well as by ICP-OES measurements to reveal calcium release or deprivation by BGN (Figure 2). Both assay types led to comparable results. BGN-containing aECM did not release a significant amount of calcium over 31 days. On the contrary, a deprivation of calcium was detected, in particular for coll BGN and coll/CS BGN, which reduced the calcium levels in the supernatant cell culture medium nearly to half at the later time points of incubation. This effect was less prominent for coll/sHA3 BGN coatings.

### 2.2. Morphology of aECM Coatings

SEM analysis of the different aECM revealed coatings with BGN decorating the surface as well as being embedded within the matrices (Figure 3A–F). The fibrillar structure of collagen was observed in all samples, while spherical BGN were observed both as discrete nanoparticles with size of 300–400 nm and as agglomerates in the micrometer size range.

### 2.3. Viscoelastic Properties of aECM Coatings

The viscoelastic properties of the different aECM variants were analyzed by oscillatory rheology during a frequency sweep at 1 Hz (Figure 4).

The applied method allowed for the analysis of a thin, equilibrium swollen aECM samples on Thermanox™ cover slips. All samples exhibited viscoelastic behavior with predominantly elastic deformation behavior, illustrated by storage moduli exceeding corresponding loss moduli. The complex viscosity of the pristine hydrogels significantly increased with the addition of a charged GAG component and with the charge density of the latter. Further, the incorporation of BGN significantly increased the complex viscosity of the aECM-based coatings. The relative increase in complex viscosity upon BGN incorporation was more pronounced for coll than for coll/CS and coll/sHA3 (>1 order of magnitude versus an increase of about 5 to 7 times). The thickness of the swollen aECM was estimated to be 150–250 µm, as force transmission could be first recorded at this distance between the plate and the coverslip.

### 2.4. Proliferation of hMSC on aECM Coatings

Proliferation of hMSC on different aECM coatings was quantified by measuring the total DNA content. Cell numbers significantly increased on coll, coll/sHA3 and, in the presence of Dex, also for coll/CS over 30 days, while cells grown on aECM containing BGN did only marginally proliferate (Figure 5 and Appendix A). Cellular growth on coll, coll/sHA3 and coll/CS was further enhanced by the presence of Dex (Figure 5C,D and Appendix A) in comparison to those without Dex (Figure 5A,B and Appendix A). In the presence of Dex the proliferation on coll/sHA3 was lower than on coll, while this was not the case in the absence of Dex. In the absence of Dex, cellular growth was lower on coll/CS than coll (Appendix A).

### 2.5. Osteogenic Differentiation of hMSC on aECM Coatings

The osteogenic differentiation was assessed via ALP activity as an early marker and calcium deposition as a late marker in osteogenic medium with and without 10^−7^ M Dex. In particular for coatings containing BGN, the presence of Dex lead to higher ALP activities compared to those without Dex (Figure 6A–D and Appendix A). Depending on the donor, maximum values were, in general, detected between day 10 and day 20. When comparing coll and coll/sHA3 without BGN, the latter led to a stronger induction of ALP activity for all four donors, irrespective of the presence or absence of Dex. For coll/CS this was only the case for donor 2 in the absence of Dex (Appendix A).

On day 10, the ALP activity on coll/sHA3 was about twice as high as on coll/CS coatings for donor 3 and 4 in the presence of Dex (Figure 6C,D and Appendix A). However, the aECM with the highest activity varied from donor to donor. In the case of donor 1, the highest values for coll/sHA3 BGN were detected on day 14. Here, the order was coll/sHA3 BGN > coll/sHA3 = coll BGN > coll, suggesting a synergistic effect of sHA3 and BGN in comparison to coll/sHA3 and coll/BGN (Figure 6A). However, unlike for the BGN-containing aECM, the ALP activity for coll/sHA3 further increased, reaching its maximum on day 28. For donor 2, the highest values were detected on day 14 for coll/CS, significantly exceeding the ALP activity of BGN-containing coatings (Appendix A and Figure 6B). Here, the presence of BGN had no superior effect compared to the bare coatings. In case of donor 3 and coll/BGN, ALP values were highest on day 20, demonstrating a clear osteogenic effect of BGN compared to coll (Figure 6C). This was also apparent for coll/sHA3 BGN and coll/CS BGN compared to coll/sHA3 and coll/CS, respectively (Appendix A). Effects for CS and sHA3 were comparable for this donor. Finally, donor 4 reached its highest ALP activity for coll/sHA3 BGN on day 10. Here, the order was coll/sHA3 BGN > coll/sHA3 = coll BGN ≥ coll/CS BGN > coll/CS = coll, again indicating a synergistic effect of sHA3 and BGN (Figure 6D; Appendix A).

The analysis of cell lysates for aECM mineralization revealed that, in the absence of Dex, substantial calcium deposition was only present on BGN-containing coatings (Figure 6E,F and Appendix A). In the case of donor 1, calcium was detected on day 28 only with no significant difference between coll BGN and coll/sHA3 BGN (Figure 6E). For donor 2, calcium deposition was already detected on day 14 and was significantly higher for coll/sHA3 BGN compared to coll BGN and coll/CS BGN on day 28 (Figure 6F and Appendix A). In the presence of Dex, calcium deposition was detected on all coatings with the highest levels on BGN-containing aECM. In the case of donor 3, the highest calcium levels were found on day 30. The order here was coll/sHA3 BGN = coll BGB > coll/CS BGN ≥ coll ≥ coll/sHA3 = coll/CS (Figure 6G and Appendix A), while at day 20, the calcium deposition on coll/sHA3 and coll/CS exceeded that on coll only, suggesting a mild synergistic effect. Finally, donor 4 had the highest calcium level for coll BGN on day 20 with the order coll BGN > coll/CS BGN ≥ coll/sHA3 BGN > coll = coll/CS = coll/sHA3 (Figure 6H and Appendix A). However, differences of BGN-containing coatings were not significant on day 30.

### 2.6. Release of Ions from aECM Coatings during Cell Culture

All BGN containing aECM showed release of silicon ions into the cell culture supernatant. After 14 days of incubation, the silicon ion release decreased in the cell-free samples, while the released concentration in the presence of cells was significantly higher compared to the cell-free samples (Figure 7). Interestingly, the release of silicon ions was slightly lower on coll/sHA3 BGN compared to coll and coll/CS coatings in the presence of cells. Calcium concentration of cell culture medium was reduced for all groups of aECM, especially at the later time points of cultivation (Appendix A).

## 3. Discussion

The aim of this study was to develop aECM surface coatings containing collagen type I, sGAG with different sugar backbone and degree of sulfation as well as BGN to assess their potential in synergistically promoting human bone formation. To this end, the physicochemical characteristics of these coatings and their effect on the proliferation and osteogenic differentiation of hMSC was investigated in vitro.

Sirius red and Toluidine blue staining revealed the presence of collagen and sGAG in the respective coatings and even distribution on tissue culture polystyrene. In contrast, in matrices with collagen as the only organic component, the collagen fibrils appeared to be less homogeneously distributed. This can be attributed to the association of sGAG with collagen fibrils shielding them from each other due to electrostatic repulsion as already suggested for the accretion of GAG-bound collagen monomers during in vitro fibrillogenesis of collagen in the presence of sGAG [30]. In line with this, it was our observation that aECM pellets from fibrillated collagen I only were rather dense and difficult to resuspend before coating, while sGAG-containing pellets appeared softer and were easier to resuspend.

The collagen content of the coatings directly after preparation was the highest for pure collagen matrices and lowest for CS-containing samples. This can in part be attributed to the preparation process in which the aECM pellet was adjusted to a final aECM concentration of 1.25 mg/mL. While in case of coll/sGAG coatings these pellets contained coll and GAG, for coll coatings it consists of the protein only, leading to a higher initial protein concentration in the coating solution. The abovementioned incomplete resuspension of the collagen only pellets might further explain the higher standard deviation for the detected collagen mass.

The initial GAG content was approximately 1.3 to 1.4 times higher for sHA3 than for CS, which is in line with the higher initial amount used for sHA3 providing the same molarity of disaccharides and thus possible binding sites for collagen. Nevertheless, in the present coatings the initial collagen-to-GAG ratio was comparable, with 86–88% collagen and 12–14% GAG. 

All aECM matrices released collagen and GAG when immersed in PBS at 37 °C, the release of GAG being much faster compared to the release of collagen. While after 13 days, irrespective of the aECM composition, there were still more than three-quarters of the initial collagen content present, the CS-containing aECM lost 84–88% of their initial CS within one day. In contrast to CS, sHA3-containing matrices lost only 35–44% within the first day with a significantly lower release of sHA3 in the presence of BGN. Previous experiments involved the dissolution of coll/GAG coatings in which GAG were mixed with collagen I already before fibrillogenesis [11]. Interestingly, the release of collagen and GAG was quite similar to the present study with less than 8% loss of collagen after 8 days compared to 50% and 83% loss of sHA3 and CS, respectively, within the first hour, suggesting that the interaction and release profiles of sGAG with collagen are not strongly influenced by the mode of sGAG incorporation. Further, the lower release of sHA3 in comparison with CS is in line with findings of a stronger interaction of high-sulfated sHA3 with collagen compared to low-sulfated sHA1 and CS [30]. In the present study, hMSC were seeded on freshly prepared aECM. Therefore, the strong sGAG release within the first day suggest an exposure of cells to 30–50 µg/cm^2^ solute GAG quantities before the first medium change, which is in contrast to previous studies were cells were seeded on aECM coatings after this initial burst release [11]. At later time points, the release on the present coatings gradually ceased with less than 5 µg/cm^2^ at day 4 and below 1 µg/cm^2^ afterwards. At day 13, 41–52 µg/cm^2^ sHA3 was still present in the coating while this was only 4–7 µg/cm^2^ CS, with BGN-containing coatings in each case having a higher GAG content. Thus, starting from day 4, the main GAG content able to affect cells, was associated with the aECM. The higher GAG content in BGN-containing coatings might be due to stabilizing hydrogen bonding between hydrated silanol groups (Si-OH) of BGN interacting with the carboxyl groups and ionic complexation of negatively charged carboxyl and sulfate groups by BGN [31]. This is in line with previous findings on a negative surface charge of collagen-based aECM coatings containing sGAG [32]. Incubation of BGN containing aECM coatings in a cell culture medium led to a calcium deprivation of the medium. It has been demonstrated before that immersion of bioactive glass in DMEM leads to crystallization of calcite and precipitation of amorphous calcium phosphate on the surface of the material [33], thereby depriving the medium from calcium ions. This is consistent with previous reports on BGN promoting mineralization of gelatin- and CS-based hydrogels and coatings [27,31,34]. Further, all BGN-containing coatings gradually released silicon ions with higher released concentrations during the first 10 days of immersion and gradually ceasing released silicon ion concentrations afterwards, indicating dissolution of the BGN in the cell culture medium. Similar results were obtained in a previous study, where pure BGN of the same composition were incubated in DMEM [18].

Continuous aECM coatings with BGN either decorating the surface or being embedded within the matrices could be revealed by SEM analysis verifying the fibrillar structure of collagen and the previously detected size of single nanoparticles with 300–400 nm [18]. However, nanoparticles also formed microsized agglomerates in the aECM coatings. The agglomeration of bioactive glass particles has been reported previously when included at higher concentrations in hydrogels made of dextran or heat-induced whey protein isolate [35,36]. Further, a certain extent of agglomeration seems to be a common side effect of the high-temperature treatment while preparing BGN via the sol–gel process and particularly relevant for BGN with smaller sizes [37]. In addition to this annealing step, the decrease in surface charge during particle synthesis is thought to be responsible for particle agglomeration.

The aECM coatings of this study exhibit elastic, gel-like properties with an increasing complex viscosity depending on the GAG component and its charge. The complex viscosity was further increased by the addition of BGN, which was more pronounced for the aECM composed of collagen only. This is in line with previous reports on bioactive nanoparticles changing the mechanical properties of hydrogels when incorporated [38]. Zhou et al., for example, also found enhanced storage and loss moduli when mesoporous bioactive glass nanoparticles were included in hybrid gelatin/oxidized CS hydrogels [31]. It has been suggested that an enhanced cross-linking degree of hydrogels via BGN is involved in this effect, e.g., due to abovementioned hydrogen bonding and complexation of released ions present in the hydrogel components [31].

In vitro cell culture revealed that hMSC numbers significantly increased on coll, coll/CS and coll/sHA3 coatings over 28 or 30 days, with the strongest proliferation on coll in the presence of Dex. In contrast, cells grown on aECM containing BGN only marginally proliferated. An enhanced cellular growth of hMSC on aECM in differentiation medium containing Dex was previously reported by Wojak-Cwik et al. [39]. Further, a slightly retarded proliferation on coll/sHA3 in comparison to collagen has been formerly shown as well, while coatings containing CS did not significantly influence the proliferation of human and murine MSC [15,39,40]. Regarding the effects of bioactive glass nanoparticles on cellular proliferation opposing findings have been reported in the literature. Zheng et al. found no cytotoxic effect on rat MSC (rMSC) for medium extracts derived from BGN preincubation, the latter having the exact same composition as used in the present study [18]; there was even a slight increase in MTT activity for cells grown in medium extracts for 24 h. Rottensteiner et al., however, reported a slightly cytotoxic effect of alginate/gelatin-hydrogels containing 45S5 glass nanoparticles after 48 h direct cultivation of hydrogel-seeded rMSC [41]. The same was found by Douglas et al. for an osteosarcoma cell line MG63 directly cultivated on gellan gum hydrogels containing sol–gel derived nanoparticles with a higher molar percentage of CaO compared to the present study [42]. Thus, all samples were incubated for 7 days in PBS to remove potentially cytotoxic substances. A slight increase in proliferation was then shown after 14 days of rMSC culture. However, at the same time a significant reduction in the mitochondrial activity was detected, suggesting that higher local concentrations of calcium ions might be responsible for the cytotoxic effects, as concentrations above 10 mM are considered cytotoxic [20]. Since, in the present study, BGN deprived the cell culture medium from calcium, an elevated calcium concentration can be excluded as reason for the retarded proliferation of the cells on BGN-containing coatings. However, low calcium concentrations have also been shown to retard proliferation of hMSC [43,44], which might be an explanation for the observed growth arrest observed in the present study. On the other hand, the silicon ions constantly released from BGN-containing coatings in the presence of cells might have counteracted this effect, since there are previous studies reporting on the stimulatory, dose-dependent effect of silicon ions on the proliferation of MSC and osteoblasts [22,25]. However, the silicon ion concentrations used in these studies were in the lower micromolar range, while in the current study 1–1.5 mM were present in cell culture media. Qazi et al. found that metabolic activity, proliferation and cell spreading of hMSC were adversely affected by higher bioactive glass concentrations [45]. Importantly, a stronger reduction in hMSC proliferation was observed when cells were grown in direct contact to two different types of bioactive glass in comparison to the exposure to higher concentrations of the respective dissolution products. Thus, our results are in line with their conclusion that cells may be less tolerant to higher concentrations of certain bioactive class compositions when physical interaction takes place (direct contact vs. indirect contact).

The presence of Dex induced higher activities of the early osteogenic marker ALP, in particular for coatings containing BGN. This is in line with previous findings by Wojak-Cwik et al. for hMSC grown on aECM-coated poly(L-lactide-co-glycolide) scaffolds [39]. Further, the stronger ALP induction found for coll/sHA3 versus coll only coatings with all four donors and the about twice as high ALP activity on day 10 for coll/sHA3 versus coll/CS for two out of three donors corroborate earlier findings by Hempel et al., Wojak-Cwik et al. and Hintze et al. [11,14,39,40]. Hence, it can be concluded that the positive effect of sHA3 in collagen coatings is independent from the association time point during or after collagen fibrillogenesis. However, synergistic effects of sHA3 and BGN on ALP activity were observed only for the cells of two donors (1 and 4). For donor 2, there was no superior effect of BGN compared to the bare coatings, while donor 3 demonstrated an osteogenic effect of BGN for coll BGN, coll/CS BGN, and coll/sHA3 BGN coatings. This demonstrates that donor variability is an important factor in assessing the effect of biomaterials and results might be at bias when looking at just one donor. BGN effects might be attributed to silicon ions, which were demonstrated to be constantly released from BGN-containing coatings in the presence of cells. Previous in vitro cell culture studies demonstrated the positive impact of silicon ions on the expression of typical osteoblast markers like ALP, osteocalcin (OCN), osteopontin (OPN), and bone sialoprotein (BSP) [19,22,25]. Moreover, the increased mechanical properties/stiffness of BGN-containing coatings, as demonstrated by oscillatory rheology, might add to the effect of released silicon ions [31]. Still, the release of silicon ions seems to be slightly diminished on coll/sHA3 BGN compared to coll BGN coatings in the presence of cells, which might hamper a stronger synergistic effect of sHA3 and BGN. Since it was previously shown that the isoelectric point of collagen-based aECM coatings decreased with increasing degree of GAG sulfation [32], this limited release from coll/sHA3 BGN is likely due to enhanced complexation of positively charged silicon ions by the enhanced negative surface charge.

The results for the mineralization as late osteogenic marker did not reflect the findings of the ALP activities in all aspects. For three donors a strong calcium deposition was already apparent as early as day 14 or 20, which is in line with previous studies on coll/GAG-based matrices, for which GAG association took place during collagen in vitro fibrillogenesis [11,14,39]. However, while the presence of BGN clearly promoted calcium deposition for all donors, there was no synergistic effect of sHA3 and BGN. Moreover, even though for donor 2, ALP activity was significantly enhanced on coll/sHA3 and coll/CS compared to all other coatings, this was not reflected by a strong calcium deposition. In fact, only cells on the BGN-containing coatings showed calcium deposition. This contrasts with previous findings where calcium deposition was increased on coll/sHA3 coatings, even in the absence of Dex [11,14,39]. As mentioned, the collagen-to-GAG ratios were comparable after 1 d incubation in PBS at 37 °C for aECM coatings prepared either by fibrillating collagen in the presence of GAG [11,14,39] or those for which collagen was fibrillated prior to non-covalent GAG association. Thus, it must be concluded that these differences in calcium deposition are related to differences in collagen structure and/or the gamma-irradiation of the latter.

Regarding the missing synergistic effects of sHA3 and BGN in the calcium assay, it can be speculated that the previously mentioned matrix mineralization even in the absence of cells might interfere with a clear readout regarding this late osteogenic marker. Here, the calcium deprivation of the medium was in particular prominent for coll BGN and coll/CS BGN and less pronounced for coll/sHA3 BGN. Thus, a stronger, cell-independent mineralization tendency of coll BGN and coll/CS BGN can be suggested. This is interesting since, e.g., CS has been implicated in initiating the formation of hydroxyapatite due to its negatively charged sulfate groups binding charged calcium and phosphate ions [46].

Our findings suggest that the composition of collagen, sGAG, and BGN in the aECM could be a promising tool in bone repair due to its potential synergistic effect on osteogenic differentiation. By further investigation of BGN composition, particle size, and GAG combination especially in in vivo studies, different fields of potential medical applications should be investigated, e.g., critical size defects, augmentation, or surface coating to further improve its clinical utilization. Since age and chronic diseases have a significant impact on bone healing, patients who belong to these groups can especially benefit from enhanced cell differentiation, which comes with accelerated calcification and consequential osseointegration, resulting in a more rapid bone healing process.

## 4. Materials and Methods

For all experiments hyaluronan (from Streptococcus *zooepidemicus*, MW = 1100 kDa; Aqua Biochem, Dessau, Germany), sulfur trioxide/dimethylformamide complex (SO_3_-DMF, purum, 97%, active SO_3_ 48%; Fluka Chemie, Buchs, Switzerland), chondroitin sulfate, from porcine trachea (CS A/C; a mixture of 70% chondroitin-4-sulfate and 30% chondroitin-6-sulfate; Kraeber & Co., GmbH, Ellerbek, Germany) and rat tail collagen type I (Corning Discovery Labware; Bedford, MA, USA) were used.

The high-sulfated HA (sHA3) was synthesized and characterized according to previous protocols [47,48]. Chondroitin sulfate (CS) was purified by dialysis and subsequent freeze-drying. Analytical data of the used polymeric GAG derivatives (Figure 8) are summarized in Table 1. 

### 4.1. Synthesis of Bioactive Glass Nanoparticles (BGN)

BGN were synthesized using a modified Stöber method reported previously [18]. The characteristics of the BGN have been reported in a previous publication [18]. BGN exhibited a size of 300–400 nm and contained 91.6S_i_O_2_-8.4CaO (mol%).

### 4.2. Preparation of aECM

The aECM was prepared in several consecutive steps: First, in vitro-fibrillogenesis of 1 mg/mL collagen type I took place in phosphate buffer (25 mM sodium dihydrogen phosphate and 5.5 mM potassium dihydrogen phosphate, pH 7.4) for 18 h at 37 °C, followed by centrifugation at 3000 rcf for 5 min at RT. The resulting pellet was washed with deionized water, centrifuged, and resuspended with either deionized water or sGAG (5 mM disaccharides in dH_2_O, which was equal to 1.4 mg CS/mg collagen and 2 mg sHA3/mg collagen) at RT by briefly vortexing. After 2 h of incubation and another centrifugation step, the samples were washed with deionized water and centrifuged again. For samples containing BGN the latter were suspended in dH_2_O (mass BGN equivalent to the initial mass of collagen) and applied to the pellet by vortexing to reach a final aECM concentration of 1.25 mg/mL. For samples without BGN, the procedure was followed accordingly but in the absence of BGN. The respective fractions were subjected to tissue culture polystyrene at 526 µg aECM/cm^2^, dried under laminar flow, and finally sterilized by gamma-irradiation at 25 kGy. Six different matrices were prepared: collagen (coll), collagen with BGN (coll BGN), collagen with CS (coll/CS), collagen with CS and BGN (coll/CS BGN), collagen with sHA3 (coll/sHA3), and collagen with sHA3 and BGN (coll/sHA3 BGN).

### 4.3. Composition and Stability of aECM Coatings

The composition and stability of aECM coatings was qualitatively and quantitatively assessed by colorimetric assays after incubation in phosphate buffered saline (PBS) or Dulbecco’s Modified Eagle’s Medium (DMEM) supplemented with 10% fetal calf serum (FCS) at 37 °C in an incubator with humidified 5% CO_2_ atmosphere for up to 31 days in the absence of cells. Supernatants as well as initial coatings were investigated by Lowry and o-phthaldialdehyde assay or DMMB for released or remaining collagen or sGAG content, respectively, as described previously [9,11]. In addition, the coatings were qualitatively analyzed for collagen and sGAG content by Sirius Red and Toluidine Blue staining as described in Picke et al. [9]. The calcium concentration in the immersion medium was determined by Fluitest^®^ Ca-CPC (Analyticon Biotechnologies AG, Germany) according to manufacturer’s instruction measuring absorbance at 590 nm as well as by inductively coupled plasma-optical emission spectrometry (ICP-OES, Plasmaquant Elite, Analytic Jena, Germany).) Ultrapure water with 2% nitric acid was used for sample preparation and all dilutions. Calibrant solutions for silicon and calcium were prepared from certified solutions (TraceCERT Merck Millipore). The use of glass equipment was avoided during sample preparation to prevent Si contamination. Disposable polypropylene tubes were used for sample collection and dilution.

### 4.4. Morphology and Viscoelastic Properties of aECM Coatings

The morphology of aECM coatings with and without BGN was assessed by scanning electron microscopy (SEM) with a Zeiss Gemini DSM 982 at 1–2 kV and a working distance of 5–6 mm. The rheological properties of equilibrium-swollen aECM and aECM/BGN were analyzed at 37 °C by oscillatory rheology using a Physica MCR-301 rheometer (Anton Paar, Graz, Austria) with an 8 mm plate geometry and a gap size of 0.05 mm. Zero gap setting was performed with an empty cover slip (Thermanox™ with Nunclon™ Delta surface treatment, Thermo Scientific™ Nunc™). A frequency sweep (0.1–10 Hz) was applied to the samples with normal force control (0.1 N). Storage modulus (G′), loss modulus (G″), and complex viscosity (η*) values were recorded with a displacement of 0.17 mrad. Data values obtained at 1 Hz were compared between samples.

### 4.5. Cell Culture on aECM Coatings

Human mesenchymal stroma cells (hMSC), isolated from bone marrow of young healthy donors (donor 1: male, 24 years; donor 2: male, 25 years; donor 3: male, 33 years; donor 4: male, 32 years) (routinely characterized and kindly provided by Prof. Bornhäuser-Medical Clinic I University Hospital Carl Gustav Carus, Dresden) were seeded at a density of 1 × 10^4^ cells/cm^2^ on aECM coated well plates and grown in a humidified 5% CO_2_ atmosphere at 37 °C. For proliferation, DMEM supplemented with 20% fetal bovine calf serum and 100 U/mL penicillin and 100 µg/mL streptomycin (P/S) was used (all Biochrom GmbH, Berlin, Germany). Four days after seeding, osteogenic differentiation was induced by changing the medium to DMEM supplemented with 10% fetal bovine calf serum, 10 mM β-glycerophosphate, 2 mM L-glutamine, 300 µM L-ascorbic acid 2-phospate, with or without 10^−7^ M Dex (all from Sigma Aldrich, Taufkirchen, Germany) and P/S. Medium was changed 2 times a week. After 7, 14, 21 or 28 days of incubation, osteogenic differentiation was analyzed.

### 4.6. Analyses of Proliferation and Osteogenic Differentiation on aECM Coatings

Cellular proliferation was quantified via determination of DNA content at different time points. Therefore, frozen samples were thawed, followed by cell lysis for 50 min with PBS containing 1% Triton X-100. During cell lysis samples were sonicated for 10 min at 80 W. Total DNA was quantified for calculation of cell number via Quantifluor assay (Promega, Madison, WI, USA) according to manufacturer´s instruction using a cellular calibration curve and measuring the fluorescence intensity (excitation: 485 nm, emission: 535 nm). ALP activity was measured using 25 µL cell lysate incubated with 125 µL ALP substrate, containing 1 mg/mL p-nitrophenyl disodium phosphate hexahydrate (Sigma-Aldrich, Germany), 0.1 M diethanolamine, 1 mM MgCl_2_, 0.1% Triton X-100 in PBS (pH 9.8), at 37 °C. p-Nitrophenol (Sigma-Aldrich; Germany) served as a calibration curve. Absorbance was determined at 405 nm. Calcium deposition was quantified in cell lysates by Fluitest^®^ Ca-CPC at 590 nm after adding 6 M HCl (ratio 1:24). ALP activity and calcium deposition were normalized to the cell number determined by DNA quantification. All measurements were performed by using an automated plate reader.

### 4.7. Statistics

Data were analyzed with OriginPro 2019 software using one-way or two-way analysis of variance (ANOVA). For multiple comparison Bonferoni and Turkey post hoc tests were applied. If not otherwise stated quadruplicates of samples were analyzed. Values for *p* < 0.05 were considered statistically significant.

## Figures and Tables

**Figure 1 ijms-22-12819-f001:**
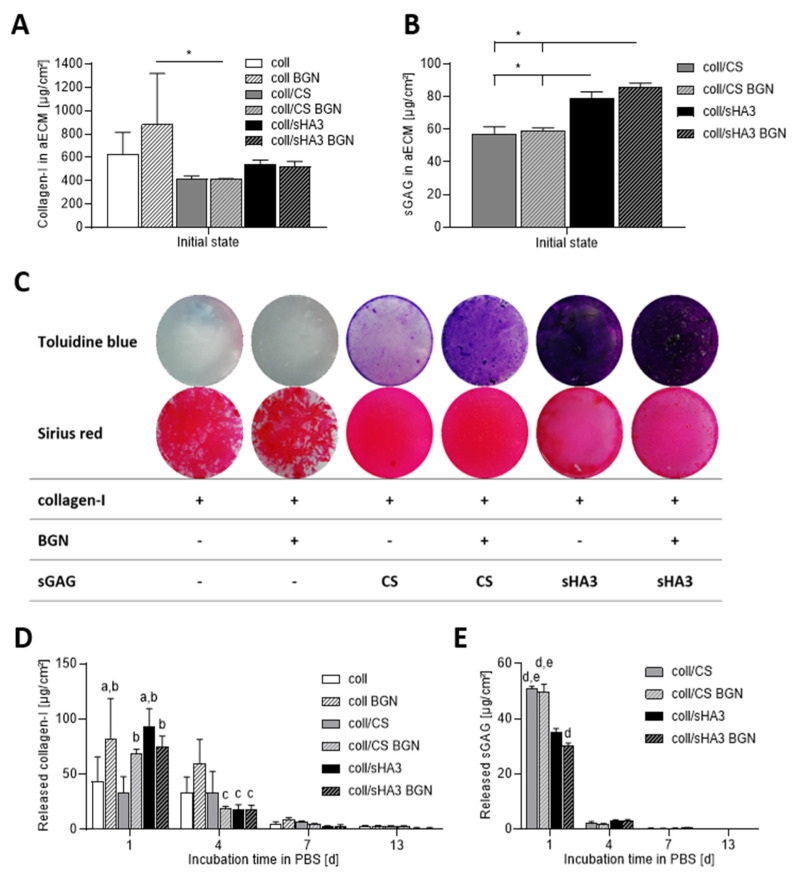
Composition, collagen, and sGAG release of aECM coatings: The amount of collagen and sGAG was determined in aECM by o-phthalaldehyde and DMMB assays after preparation (day 0). Collagen content in (**A**), sGAG content in (**B**). Qualitative analyses of aECM composition after preparation is shown in (**C**) using Sirius red and Toluidine blue staining for detecting collagen and sGAG, respectively. The amount of collagen (**D**) and sGAG (**E**) was determined in supernatants by Lowry and DMMB assays over 13 days at 37 °C in PBS. One-way ANOVA: * *p* < 0.05 (**A**,**B**). Two-way ANOVA: *p* < 0.05 was considered statistical different to coll 1 d (a), coll/CS 1 d (b), coll BGN 4 d (c), coll sHA3 1 d (d), and coll/sHA3 BGN 1 d (e).

**Figure 2 ijms-22-12819-f002:**
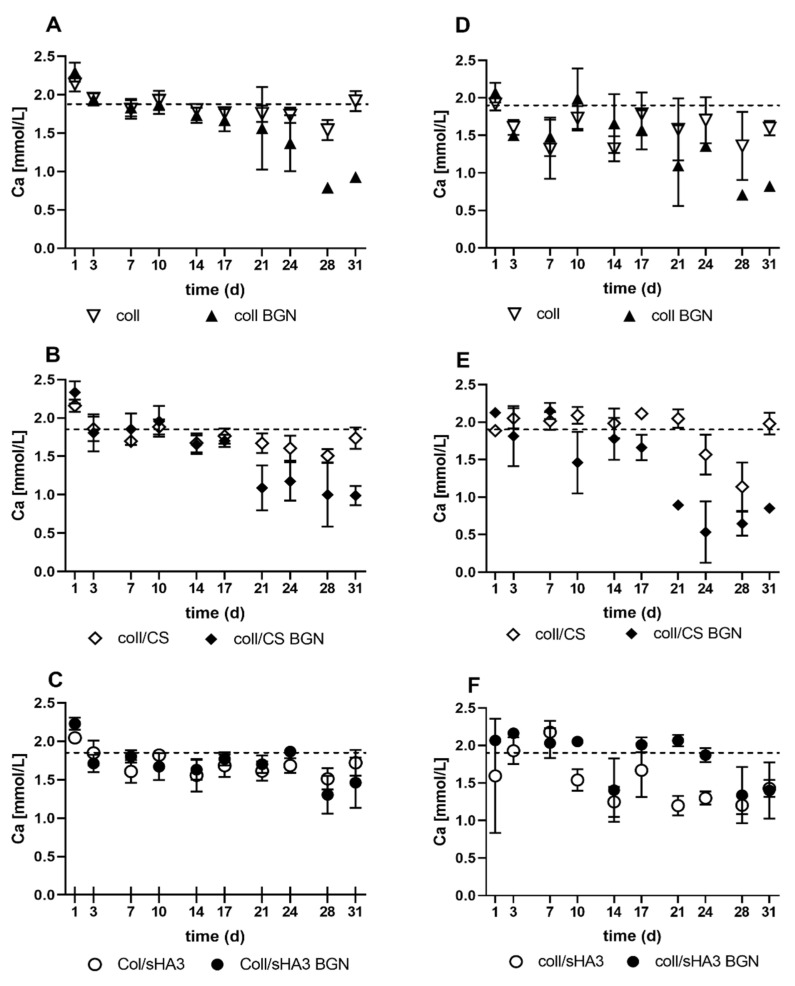
Calcium release from aECM: The calcium concentration was determined from supernatants of aECM incubated over 31 days at 37 °C in DMEM medium including 10% FCS and osteogenic supplements. Four different samples were analyzed per group and time point. (**A**–**C**): Calcium concentration as quantified by colorimetric analysis, (**D**–**F**) calcium content as quantified by ICP OES (see experimental section for further details). The dotted line represents the calcium concentration in the medium.

**Figure 3 ijms-22-12819-f003:**
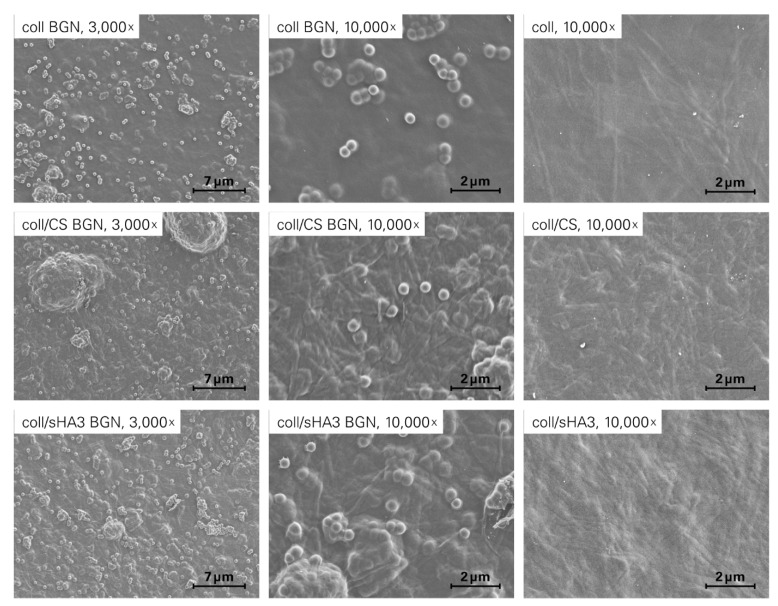
SEM micrographs of different aECM. Micrographs were taken from the initial state of different aECM coatings at 3000× and 10,000× magnification.

**Figure 4 ijms-22-12819-f004:**
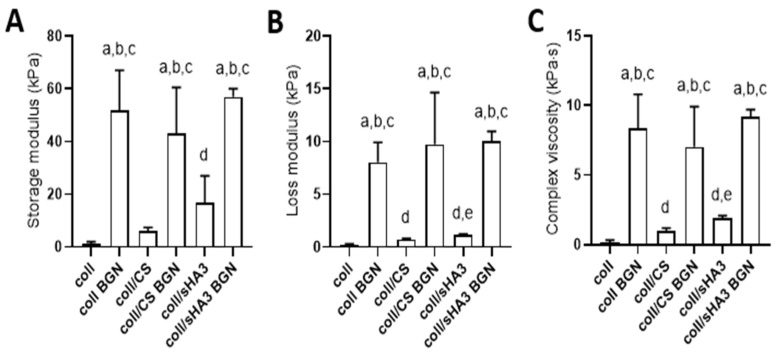
Viscoelastic characteristics of aECM coatings: Rheological measurements were performed at 1 Hz. Storage modulus (**A**), loss modulus (**B**), complex viscosity (**C**). Statistic: *p* < 0.05 was considered significant. One-way ANOVA over all data with significant differences to coll (a), coll/CS (b), or coll/sHA3 (c). One-way ANOVA for data without BGN with significant difference to coll (d) and coll/CS (e).

**Figure 5 ijms-22-12819-f005:**
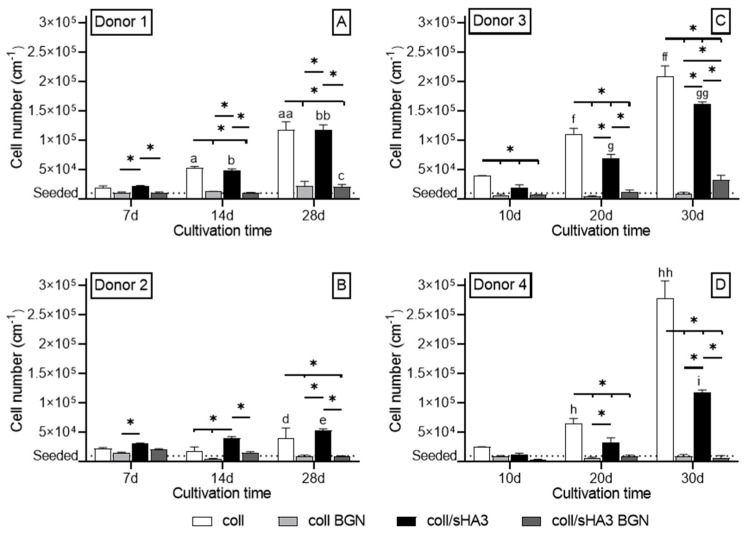
Cell proliferation. Cellular growth of hMSC as determined by analysis of the DNA content after addition of osteogenic supplements: (**A**/**B**). hMSC cultivated w/o Dex (Donor 1/2) and (**C**/**D**). hMSC cultivated with Dex (Donor 3/4); n = 4. Statistics: 2-way ANOVA * *p* < 0.05 was considered significant. The following data also showed significant differences on *p* < 0.05 considering the respective graph. Data significantly different to a—coll 7d; aa—coll 7d and 14d; b—coll/sHA3 7d; bb—coll/sHA3 7d and 14d; c—coll/sHA3 BGN 7d; d—coll 7d and 14d; e—coll/sHA3 7d; f—coll 10d; ff—coll 10d and 20d; g—coll/sHA3 10d; gg—coll/sHA3 10d and 20d; h—coll 10d; hh—coll 10d and 20d; i—coll/sHA3 10d and 20d.

**Figure 6 ijms-22-12819-f006:**
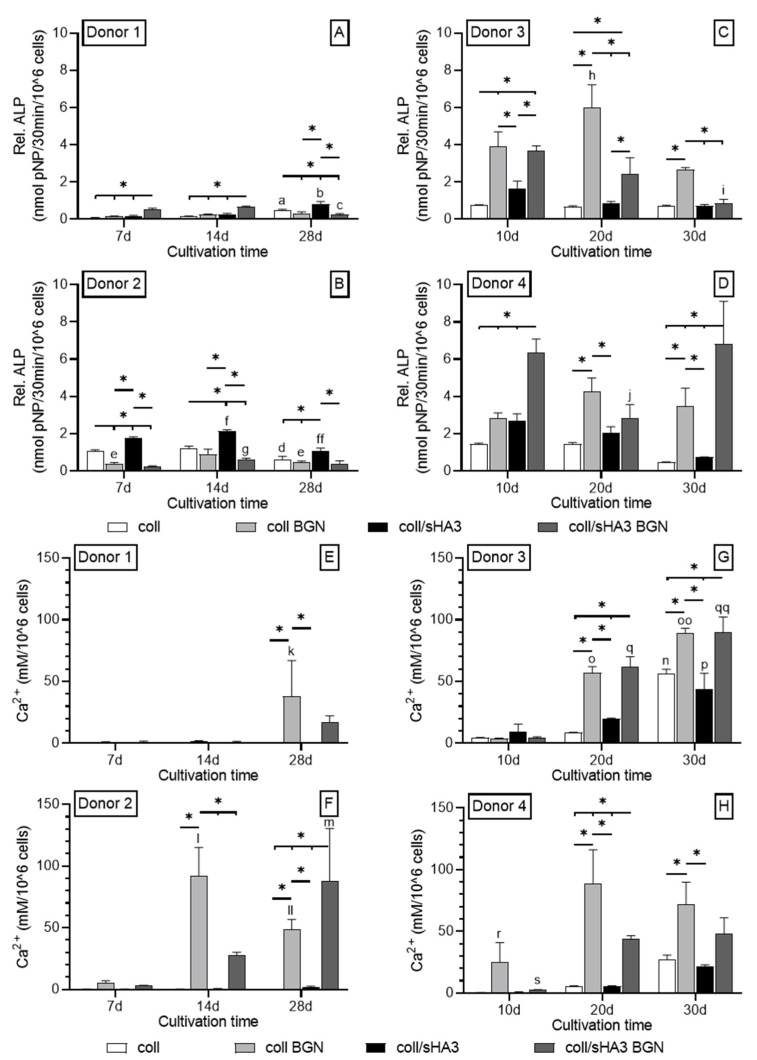
ALP activity and calcium deposition: Relative ALP activity (**A**–**D**) and calcium deposition (**E**–**H**) of hMSC after addition of osteogenic supplements. hMSC on aECM coatings w/o Dex (Donor 1/2) and coatings with Dex (Donor 3/4); n = 4. Statistics: 2-way ANOVA * *p* < 0.05 was considered significant. The following data also showed significant differences on *p* < 0.05 considering the respective graph. Data significant different to a—coll 7d and 14d; b—coll/sHA3 7d and 14d; c—coll/sHA3 BGN 7d and 14d; d—coll 7d and 14d; e—coll BGN 14d; f—coll/sHA3 7d; ff—coll/sHA3 7d and 14d; g—coll/sHA3 BGN 7d; h—coll BGN 10d and 30d; i—coll/sHA3 BGN 10d and 20d; j—coll/sHA3 BGN 10d and 30d; k—coll BGN 7d and 14d; l—coll BGN 7d; ll—coll BGN 7d and 14d; m—coll/sHA3 BGN 7d and 14d; n—coll 10d and 20d; o—coll BGN 10d; oo—coll BGN 10d and 20d; p—coll/sHA3 10d and 20d; q—coll/sHA3 BGN 10d; qq—coll/sHA3 BGN 10d and 20d; r—coll BGN 20d and 30d; s—coll/sHA3 BGN 20d and 30d.

**Figure 7 ijms-22-12819-f007:**
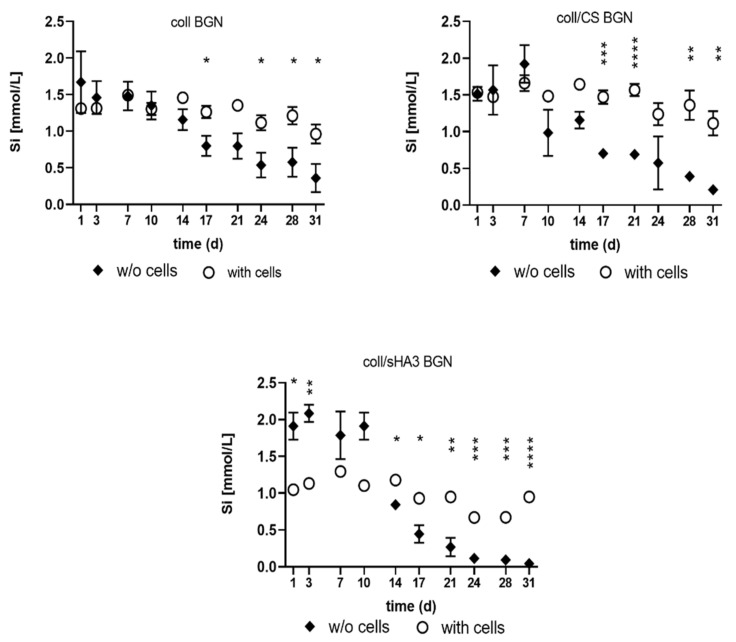
Silicon ion concentration after incubation of the BGN-containing coatings with cell culture medium (DMEM, 10% FCS, plus osteogenic supplements) in the presence and absence of cells. n = 4, average +/− standard deviation, two-way ANOVA followed by Sidak’s multiple comparisons revealed significant differences between cell-containing and cell-free samples as follows: * *p* < 0.05, ** *p* < 0.01, *** *p* < 0.001, **** *p* < 0.0001.

**Figure 8 ijms-22-12819-f008:**
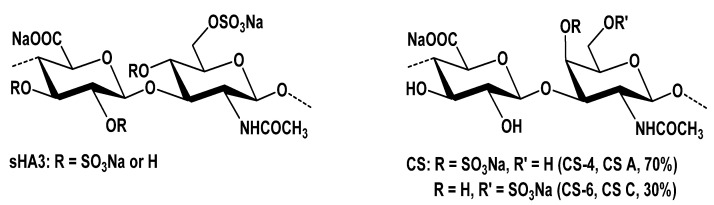
Polymeric GAG derivatives.

**Table 1 ijms-22-12819-t001:** Characteristics of GAG.

Derivative	DS	M_n_ (Da)	M_w_ (Da)	PD
CS	0.8	17 700 (40 700)	21 600 (61 800)	1.5
sHA3	3.1	34 800 (46 800)	51 300 (80 800)	1.7

DS (degree of sulfation): average number of sulfate groups per repeating disaccharide unit; M_n_ (number-average molecular weight) and M_w_ (weight-average molecular weight): analyzed by gel permeation chromatography (GPC). Values were determined with laser light scattering detection and refraction index (RI) detection (in parentheses); PD (polydispersity index): detected by GPC; values were calculated from RI detection.

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
