# Peer review of "Artificial Extracellular Matrices Containing Bioactive Glass Nanoparticles Promote Osteogenic Differentiation in Human Mesenchymal Stem Cells"

_ijms, 2021, doi:10.3390/ijms222312819_

Round 1

Reviewer 1 Report

The present study aims to evaluate the osteogenic capacity of human mesenchymal stem cells in contact with a collagen / sulfated glycosaminoglycan based surface containing bioactive glass nanoparticles. The methods used are suitable for obtaining relevant results. The only problem for which I did not find results or explanation in the description of the materials and the method refers to the four human cell lines. The authors state that they are human mesenchymal cells. Were primary lines or stabilized lines used? How has their origin been demonstrated? How has their multipotent character been demonstrated? Have they been immunophenotypically analyzed? Do they comply with the rules imposed by the International Society for Cellular Therapy?

Reviewer 2 Report

This manuscript describes the investigation of different bio-inorganic coatings for promoting the osteogenic differentiation of human mesenchymal stroma cells. The study is well carried out and manuscript of an excellent quality. A weakness of the manuscript lies in the limited characterisation of the coatings properties:

More specifically, the drying approach used to produce the ECM coatings may lead to strong variability in and across samples, including composition and matrix alignment gradients in X/Y and Z dimensions. Also the coatings should be further characterizsed to understand in more details the parameters into play. For example, please give an indication of the coating thickness, did this thickness influence the release profiles? Were the BGNs homogeneously dispersed inside the matrix, or rather sedimented on the substrate? 

Some silicon concentration measurements are discussed, but too few details are given related to the measurement techniques used. In particular if silicon concentrations are to be measured accurately, all preparations should be carried out in glass-free labware, since storage of aqueous solutions in glass containers typically lead already to measurable levels of soluble silicon. Moreover silicon is already present in certain cell culture products. Could you please comment on this?

A few typos:

L44: “Consideration the limitations of traditional therapies”: Word(s) are probably missing.

L148: “the analysis of a thin, equilibrium swollen aECM samples“: Please check grammar.
